# Defect Detection of MEMS Based on Data Augmentation, WGAN-DIV-DC, and a YOLOv5 Model

**DOI:** 10.3390/s22239400

**Published:** 2022-12-02

**Authors:** Zhenman Shi, Mei Sang, Yaokang Huang, Lun Xing, Tiegen Liu

**Affiliations:** 1School of Precision Instrument and Opto-Electronics Engineering, Tianjin University, Tianjin 300072, China; 2Key Laboratory of Opto-Electronics Information Technology, Ministry of Education, Tianjin 300072, China

**Keywords:** defect detection, MEMS, data augmentation, GAN, YOLOv5

## Abstract

Surface defect detection of micro-electromechanical system (MEMS) acoustic thin film plays a crucial role in MEMS device inspection and quality control. The performances of deep learning object detection models are significantly affected by the number of samples in the training dataset. However, it is difficult to collect enough defect samples during production. In this paper, an improved YOLOv5 model was used to detect MEMS defects in real time. Mosaic and one more prediction head were added into the YOLOv5 baseline model to improve the feature extraction capability. Moreover, Wasserstein divergence for generative adversarial networks with deep convolutional structure (WGAN-DIV-DC) was proposed to expand the number of defect samples and to make the training samples more diverse, which improved the detection accuracy of the YOLOv5 model. The optimal detection model achieved 0.901 *mAP*, 0.856 F1 score, and a real-time speed of 75.1 FPS. As compared with the baseline model trained using a non-augmented dataset, the *mAP* and F1 score of the optimal detection model increased by 8.16% and 6.73%, respectively. This defect detection model would provide significant convenience during MEMS production.

## 1. Introduction

Micro- and nanoscale devices have been applied in many areas [1], especially micro-electromechanical system (MEMS) devices. They have many advantages such as low cost, small size, and the capability to integrate with on-chip circuits [2]. However, MEMS surface defects are inevitable during production, which affect the function of the device. The real-time detection of MEMS surface defects is crucial but challenging [3]. On the one hand, small size defects may lead to low detection accuracy. An object detection deep learning network with better feature extraction performance is needed. On the other hand, a deep learning network requires sufficient training samples for learning to avoid overfitting [4]. It is difficult to collect enough and diverse defect samples from the products in practice [5].

Thus, in this work, we present a methodology to deal with these problems. This method that is based on data augmentation effectively solves the problem that an insufficient original dataset affects the training of a detection model, which would reduce the reliance on dataset collection during an industrial fabrication process. In addition, the methodology uses a single-stage detection model. Smaller size defects will be detected on the production line more accurately and in real time.

### 1.1. Related Works

In this section, we provide a brief overview of previous studies related to data augmentation with GANs, and then we discuss the relevant work on object detection with single-stage and two-stage models.

Data augmentation is widely used to avoid overfitting. Conventional data augmentation methods, such as flipping, rotation, and color space transformations, can only change the angle, position, or pixel value of the defects but not the shapes [6]. There may be significant limitations on defect detection, because models cannot recognize the defects in shapes not contained in training sets. A generative adversarial network (GAN) can address this problem by generating new samples based on the distribution of an existing dataset [7]. Ssu-Han Chen et al. [8] proposed a method for wafer die defect detection by combining a GAN and a modified you only look once version 3 (YOLOv3). The experiments showed that the GAN could enhance the diversity of defects, which improved the average precision by 7.33%. However, a traditional GAN may face a mode collapse problem (i.e., the image generated by the generator lacks diversity). Different GAN variants have been proposed to address this problem [9,10]. Wasserstein GAN (WGAN) [11] effectively avoided mode collapse by introducing Earth-Mover (EM) distance to define the loss function. Yawen Xiao et al. [12] proposed a deep learning-based WGAN approach to solve the imbalanced learning problem in cancer gene expression data. The results showed that, as compared with random oversampling, the synthetic minority oversampling technique, and GAN, the classifier with the proposed WGAN model provided better prediction performance in cancer diagnosis. 

Object detection in industrial production has received a lot of attention in recent years. Ghalia Tello et al. [13] proposed a deep-structured machine learning (ML) approach to identify and to classify single defects and mixed defects on semiconductor wafers. The research improved their previous randomized general regression network (RGRN), by incorporating an information gain (IG)-based splitter and deep-structured ML. The results showed this combination achieved an overall detection accuracy of 86.17%. Amin Amini et al. [14] developed an automated defects recognition (ADR) system using a unique plenoptic camera, which could detect surface defects of MEMS wafers based on a Faster R-CNN Inception V2 COCO model. The results showed an F1 score of 0.81 U on average, with a processing time of 18 s. Nonetheless, with the improvement of detection algorithms, a deep learning detection network with a good balance between high accuracy and fast speed is needed in industrial environments. The YOLO series [15,16,17] has been one of the most popular detection algorithms in the industrial manufacturing field because of its high detection accuracy and fast speed. Ishak Pacal et al. [18] presented an improved YOLOv3 by integrating a cross-stage partial network (CSPNet) for real-time automatic polyp detection. The proposed method and the use of transfer learning with negative samples significantly increased the real-time polyp detection performance. 

### 1.2. Paper Contribution

In this paper, we aimed to train a real-time detection model based on YOLOv5 to detect MEMS surface defects more accurately and efficiently. 

First, a MEMS defect dataset was expanded with an improved GAN, i.e., Wasserstein divergence for generative adversarial networks with deep convolutional structure (WGAN-DIV-DC). To improve the quality of the generated images, different network structures were applied to the generator and discriminator. 

Secondly, in order to improve the detection performance of the YOLOv5 baseline model, Mosaic and one more prediction head were added into the baseline model for comparative experiments. The optimal model trained by using an augmented dataset performed well both in accuracy and speed during MEMS surface defect detection.

## 2. Method

### 2.1. Dataset Collection

The defect dataset was collected from a MEMS wafer containing thousands of MEMS cells. The data acquisition system consisted of three parts, as shown in Figure 1: an area array charge-coupled device (CCD) camera, a microscope, and a MEMS wafer. The microscope was equipped with a 10× eyepiece and a 20× objective lens. By moving the stage in the x and y directions, every MEMS cell could be captured by the camera. Finally, 1200 MEMS surface defect images, with a size of 1600 × 1600, were collected from over 10,000 MEMS cells.

### 2.2. Data Augmentation

It was noticed during data collection that only a few images could be used as training samples. In this step, data augmentation based on generative adversarial networks was applied to the original dataset.

A GAN training strategy defines a minimax two-player game between a generator network (*G*) and a discriminator network (*D*). The generator takes a random noise as input and outputs a generated image. The discriminator distinguishes the difference between the generated sample and the real sample and rejects the fake one. As shown in Figure 2, the two networks improve their performance by competing with each other.

During GAN training, the *G* and *D* are trained alternately, constantly optimizing the following minimax value function [7]:(1)$$\underset{G}{\min}\underset{D}{\max}V(D,G)=E_{x\sim r(x)}[\log D(x)]+E_{z\sim g(z)}[\log(1-D(G(z)))]$$
where r(x) is the distribution of the real data and g(z) is the distribution of a noise. The value of V(D,G) is minimized by training *G*, and then is maximized by training *D*. During training, the binary cross-entropy loss (BCEL) is used to define the loss function of the generator (*L_G_*) and discriminator(*L_D_*):(2)$$\begin{array}{l} L_{G}=BCEL(G(z),a)=-w[a\log G(z)+(1-a)\log(1-G(z))]\\ L_{D}=\frac{1}{2}[BCEL(D(x),b)+BCEL(D(G(z)),b)] \end{array}$$

The generator and discriminator are both multilayer perceptrons (MLP), consisting of linear connection layers, one-dimensional normalization layers, and activation layers, as shown in Figure 3a.

#### 2.2.1. Deep Convolutional Generative Adversarial Network

In the DCGAN [19], all convolutional layers were used to reduce the parameters in Figure 3b. In the generator, ReLU was used for all activation layers except for the output, which used Tanh. In the discriminator, LeakyReLU was used for all activation layers to prevent gradient sparsity. In addition, pooling layers were replaced by convolution layers (stride = 2) to prevent losing too many features.

#### 2.2.2. Wasserstein Generative Adversarial Network

The WGAN introduced Earth-Mover (EM) distance to define the loss function of the discriminator and generator, and the EM distance (*W* distance) is defined as follows:(3)$$W(r(x),g(x))=\underset{{\left\|f_{w}\right\|}_{L}\leq 1}{\sup}E_{x\sim P_{r}}[f_{w}(x)]-E_{x\sim P_{\theta }}[f_{w}(x)]$$
where *W* means the minimum cost required to move one data distribution to another; $E_{x\sim P_{r}}[f_{w}(x)]$ and $E_{x\sim P_{\theta }}[f_{w}(x)]$ indicate the real image and the generated data distribution mapped by $f_{w}(x)$, respectively; and $f_{w}$ can be considered to be a discriminator network with parameter w. $f_{w}$ should satisfy the following condition [11]:(4)$$\begin{array}{l} {\left\|f_{w}\right\|}_{L}=\underset{x\neq y}{\max}\frac{\left|f_{w}(x)-f_{w}(y)\right|}{\left\|x-y\right\|}\leq 1\\ \left|f_{w}(x)-f_{w}(y)\right|\leq \left\|x-y\right\| \end{array}$$
where ${\left\|f_{w}\right\|}_{L}$ denotes the Lipschitz norm of $f_{w}$. To enforce the Lipschitz constraint, the loss function can be defined in the following ways: weight clipping, gradient penalty [20], and Wasserstein divergence [21], corresponding to three models, i.e., WGAN, WGAN-GP, and WGAN-DIV, respectively.

WGAN limits the discriminator parameters to [−c,c] by clipping weight directly, and the loss function of the generator and discriminator can be calculated by using Equation (5):(5)$$\begin{array}{l} L_{G}=-\left[E_{z\sim g(z)}[D(G(z))]\right]\\ L_{WGAN\_D}=E_{z\sim g(z)}[D(G(z))]-E_{x\sim r(x)}D(x) \end{array}$$

WGAN-GP makes the model more stable by penalizing the gradient of the discriminator.
(6)$$L_{WAGN-GP\_D}=E_{z\sim g(z)}[D(G(z))]-E_{x\sim r(x)}D(x)+\lambda E_{\hat{x}\sim p(\hat{x})}[{\left(\left\|\nabla_{\hat{x}}D(\hat{x})\right\|-1\right)}^{2}]$$
where $\hat{x}=\varepsilon x+(1-\varepsilon )G(z)$, λ is a hyperparameter to be determined, ε is a random number ranging from 0 to 1. As the input should be continuous considering Lipschitz constraint, the input data are made to be distributed evenly in the whole space by interpolating between the generated image and the real image.

Wasserstein divergence was introduced into WGAN to define the loss function, which does not require the 1-Lipschitz constraint. The loss function of the discriminator in WGAN-DIV is:(7)$$L_{WAGN-DIV\_D}=E_{z\sim g(z)}[D(G(z))]-E_{x\sim r(x)}D(x)+kE_{\hat{x}\sim p(\hat{x})}[{\left(\left\|\nabla_{\hat{x}}D(\hat{x})\right\|\right)}^{p}]$$
where *k* and *p* are hyperparameters. The experimental results [21] show that the model had the best performance when *k* = 2 and *p* = 6.

In this study, inspired by DCGAN, the MLP structure of WGAN was replaced with a deep convolution network (DC). As shown in Figure 3c, ReLU activation in the generator was replaced with LeakyReLU to obtain a better performance. In addition, the final Sigmoid activation and batch normalization layer were removed to slow down the convergence speed of the discriminator and to optimize the generator stably. Thus, we obtained WGAN-DC, WGAN-GP-DC, and WGAN-DIV-DC. The validity of the models were proven through the experiments described in Section 3.1.

### 2.3. Yolov5 Algorithm

As introduced in the previous sections, a YOLOv5 model was used to detect MEMS surface defects. The YOLOv5 network structure mainly consisted of four parts: model input, backbone, neck, and prediction. Figure 4 shows the baseline model of this experiment. Mixup [22], a straightforward data augmentation principle, constructed a new sample by linearly interpolating two random samples and their labels from the training set. Mixup increased the robustness to adversarial samples and greatly improved the generalization of the model.

The backbone network included a Convolution-Batchnorm-SiLU module (CBS), C3_n, and spatial pyramid pooling (SPP). The CBS is the most basic module in a backbone network, consisting of a two-dimensional convolutional layer, two-dimensional batch normalization layer, and SiLU activation function. The CBS2, whose stride equaled 2, performed the downsampling operation. The bottleneck module was inspired by the idea of CSPNet [23]. Only half of the feature channels go through the CBS modules. SPP executes maximum pooling with kernel sizes 5 × 5, 9 × 9, and 13 × 13, and SPP concatenates the feature maps to avoid computing the convolutional features repeatedly.

The neck module, a feature fusion part, combines a feature pyramid network (FPN) and a path aggregation network (PAN). FPN [24] extracts the features through a top-down architecture with lateral connections, while PAN [25] transmits features in a bottom-up pyramid. The two structures showed significant improvement in feature extraction.

Three feature maps, with sizes of 20 × 20, 40 × 40, and 80 × 80, were used as three prediction layers to detect objects of different sizes. Every prediction layer outputs the corresponding prediction head, and finally works out the prediction bounding box and class. The generalized intersection-over-union (*GIoU*) [26] loss function can solve the problem of non-overlapping bounding boxes during training. As shown in Figure 5, box *C* is the minimum rectangle area including *A* and *B*. *GIoU* and the loss function were calculated by using Equation (11):(8)$$\begin{array}{l} GIoU=IoU-\frac{C-A\cup B}{C}\\ L_{GIoU}=1-GIoU \end{array}$$
where *IoU* (intersection-over-union) is the intersection ratio of the prediction box *A* and original label box *B*.

### 2.4. Modules for Comparative Experiments

In this paper, the following three modules were introduced into the baseline model to improve the detection accuracy of the model. First, as compared with Mixup, Mosaic [27] selects four samples from the training set every time for random scaling, regional clipping, disorderly arrangement, and splicing into a new image, as shown in Figure 6. Mosaic can increase the diversity of the dataset and can improve the robustness of the model.

In the backbone network, spatial pyramid pooling fast (SPPF), an improved version of SPP, was applied to speed up interference. As shown in Figure 7, SPPF serially executed the maximum pooling with kernel size 5 × 5 and fused the features by concatenation, followed by a CBS module to adjust the output channels.

In the prediction network, one more prediction head was used to detect different size objects, especially for small MEMS defects. As shown in Figure 7, the feature map was expanded to 160 × 160 by upsampling on 19 layers, and then concatenated with the 2nd layer. Four different size feature maps were used to predict bounding box and class, to enhance the feature extraction ability of the network. 

## 3. Experimental Results and Discussions

### 3.1. Data Augmentation

Considering that most of the defects only occupied a small area of the original images, defects with a size of 64 × 64 were picked up by sliding windows. A total of 640 selected defect images were used to train the GAN models. According to existing experiments, Adam optimization [21] with a learning rate of 0.0002 was used to update G and D. In order to get better convergence results, the batch size was set to 64 and the training epoch was set to 2700.

Fréchet inception distance (FID) [28] is commonly used to describe the performance of GAN models. The pretrained Inception V3 was used to propagate all real and generated images, and the last pooling layer was used as the coding layer. For this coding layer, the mean and the covariance of the real and generated images were calculated, respectively. When the real images and generated images are assumed to follow a Gaussian distribution, the difference of two Gaussians is measured by FID, which is defined as:(9)$$FID(x,g)={\left\|m_{x}-m_{g}\right\|}_{2}^{2}+Tr(C_{x}+C_{g}-2{\left(C_{x}C_{g}\right)}^{1/2})$$
where $m_{x},C_{x}$ are the mean value and covariance of the real images, respectively; $m_{g},C_{g}$ are the mean and covariance matrix of the generated images, respectively. The lower FID indicates that the generated images are more similar to real images. 

As shown in Figure 8a, the FID of DCGAN decreased rapidly in the early stage, and the fluctuation was small, indicating that the training of DCGAN was stable. The convergence of DCGAN training came at around 1000 epochs, and the lowest FID was 43.48. WGAN limited the parameters to [−0.01, 0.01] by clipping weights directly, which could easily lead to weight binarization and fall into a local extreme point. The FID of WGAN gradually converged at 1800 epochs, and the lowest FID was 54.47.

Figure 8b shows the training progress of WGAN and WGAN-DC. The FID of WGAN-DC dropped rapidly and kept a low level with small fluctuations, which indicated that WGAN-DC training was more stable. The lowest FID was 39.55, which meant the quality of the generated samples was better.

As shown in Figure 9, the training process of WGAN with DC structure was more stable, converged faster, and reached a much lower FID than WGAN with MLP structure. The lowest FID values of WGAN-GP-DC and WGAN-DIV-DC were 33.75 and 29.44, respectively. The lowest FID values of the models in Table 1 show that WGAN-DIV-DC outperformed the compared methods. 

The improved model WGAN-DIV-DC was used to expand the MEMS dataset, and 15,000 defect samples were generated. Figure 10 shows some of the generated defect samples. The generated defects were randomly pasted in defect-free MEMS images, and the number of defects on each image ranged from one to five. Finally, the original 200 training images was expanded to 4000 images. One of the synthetic training images is shown in Figure 11.

### 3.2. Detection Performance Comparison

#### 3.2.1. Training Setting

After data augmentation, the number of images was expanded to 5000, consisting of 1200 real images and 3800 generated images. As shown in Table 2, according to 8:1:1, the training set contained all the generated images and 200 real images, while the validation set and test set only contained 500 real images, respectively. 

Our experimental environment is shown in Table 3. The model training epoch was set to 3000, and the batch size was set to 16. The initial learning rate was 0.01. We trained this model only on a MEMS dataset from scratch without using pretrained weights.

#### 3.2.2. Model Evaluation Metrics

Some metrics were used to evaluate the performance of the detection models: precision (*P*), recall (*R*), mean average precision (*mAP*), F1 score, and detection speed. The detection speed can be measured by the number of images detected per second (FPS). 

In the dataset, normal samples are positive (*P’*) and defect samples are negative (*N*). The model predicts whether input samples are positive or negative, which is true (*T*) or false (*F*). Thus, predicted results can be described by true positive (*TP’*), true negative (*TN*), false positive (*FP’*), and false negative (*FN*). Precision and recall are calculated with the following equations: (10)$$\begin{array}{l} precision=\frac{TP^{\prime }}{TP^{\prime }\;+\;FP^{\prime }}\\ recall=\frac{TP^{\prime }}{TP^{\prime }\;+\;FN} \end{array}$$

It is common to use the F1 score and *mAP* to comprehensively measure the accuracy of a model. The harmonic mean of the precision and recall scores is the *F*1 score, as defined in Equation (11):(11)$$F1=\frac{2}{\frac{1}{P}+\frac{1}{R}}$$

The *P*-*R* curve is plotted with *R* as x axis and *P* as y axis. The area under the *P*-*R* curve (*AP*) and *mAP* are calculated using Equation (12):(12)$$\begin{array}{l} AP=\int_{0}^{1}P(R)dR\\ mAP=\frac{\sum_{i=1}^{N}AP_{i}}{N} \end{array}$$
where *mAP* is the average *AP* of all categories (*N* classes). The closer the *mAP* and F1 score get to one, the better performance of the model is.

#### 3.2.3. Training Results and Discussions

Figure 12 shows the influence of data augmentation on the baseline model. If the metrics did not increase, the model training terminated in advance. The model started to converge at around 600 epochs trained by using a non-augmented dataset, while after data augmentation, the model training started to converge at around 270 epochs, and terminated in advance at about 330 epochs. The detection accuracy of the models was greatly improved, as shown in Table 4.

Additionally, the inference time was tested. Table 5 presents a comparison of the inference time with different models. The detection speed of the model trained with SPPF was not significantly improved, so SPP was not replaced by SPPF finally.

It can be seen from Table 4 that the detection *mAP* of the optimal model in this paper for MEMS surface defects is 0.901 and the F1 score is 0.856. Data augmentation and Mosaic greatly improve the accuracy of the detection model. In addition, one more prediction head added one more feature map, which performed better to detect MEMS defects. Some randomly selected detection results of the optimal model are shown in Figure 13, where the green is the label box and the red is the prediction box.

In order to evaluate the overall success of our research, it was essential to provide a comparison against similar studies conducted in recent years. Table 6 presents a comparison of our method with those in [14,29].

The objective in three cases was surface defects. The authors of [14], based on a two-stage detector, identified no better balance between detection accuracy and speed, while [29] based on a one-stage detector, they reported better performance in detection accuracy and speed. Although our method had a lower *mAP* than [29], the difference was minimal and around 2.4%. In addition, our results presented a faster calculation speed. Thus, our method provided a more accurate and efficient defect detection model for MEMS production.

## 4. Conclusions

In conclusion, an improved generative model WGAN-DIV-DC and an optimal detection model were proposed to detect MEMS surface defects, which addressed the challenges of real-time defect detection during the fabrication process. First, WGAN-DIV-DC generated more diverse defect samples to expand the original dataset. Secondly, the YOLOv5 baseline model was optimized by introducing Mosaic and one more prediction head. The comparative experimental results showed that the optimal model achieved the highest *mAP* of 0.901 and F1 score of 0.856. As compared with the baseline model, the optimal model trained by using the augmented dataset performed better. This proposed framework has the potential to be used in other similar surface defect detection scenarios, which could reduce the reliance on original data collection. In addition, the visual detection method would significantly improve production efficiency and product quality. 

Although the experimental results have shown the feasibility of the proposed method for MEMS surface defect detection, we plan to explore different networks and parameters for model optimization in future works. 

## Figures and Tables

**Figure 1 sensors-22-09400-f001:**
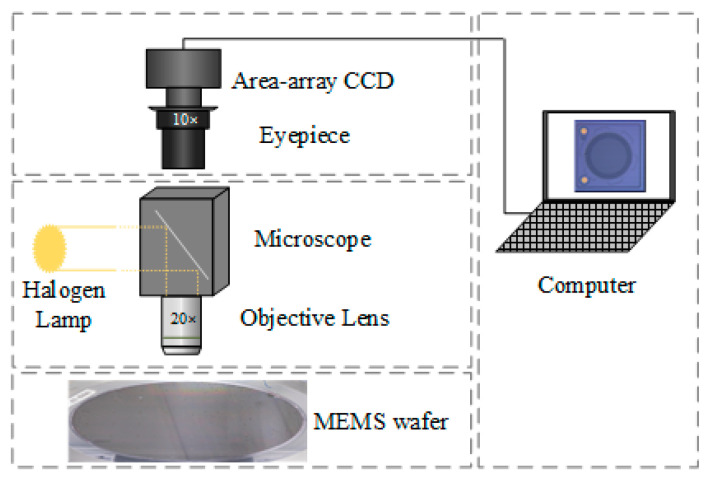
Data acquisition system.

**Figure 2 sensors-22-09400-f002:**
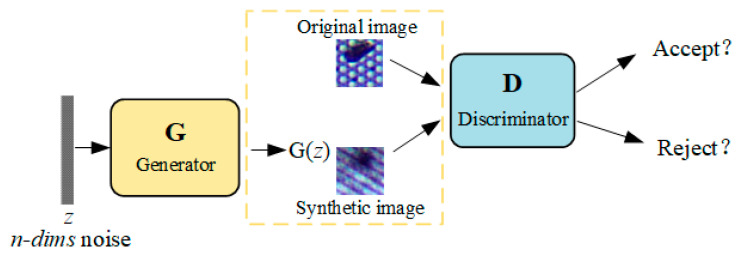
GAN diagram.

**Figure 3 sensors-22-09400-f003:**
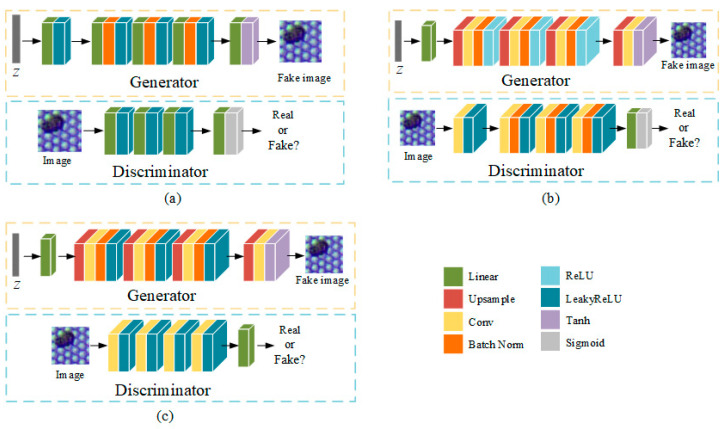
Different network structures of GAN variants: (**a**) GAN or WGAN with MLP; (**b**) deep convolutional GAN (DCGAN); (**c**) WGAN with deep convolutional structure (DC).

**Figure 4 sensors-22-09400-f004:**
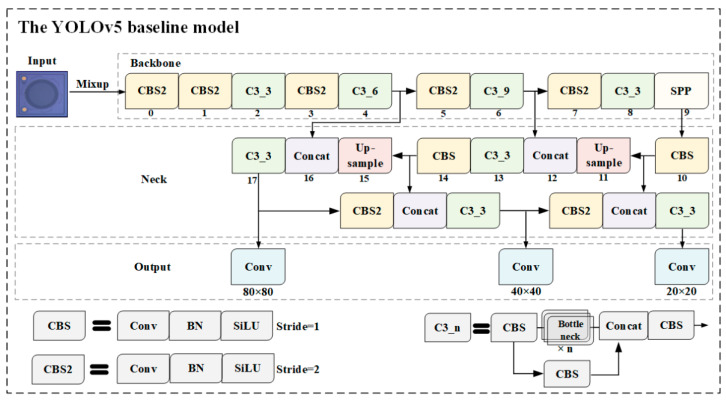
The network structure of the YOLOv5 baseline model.

**Figure 5 sensors-22-09400-f005:**
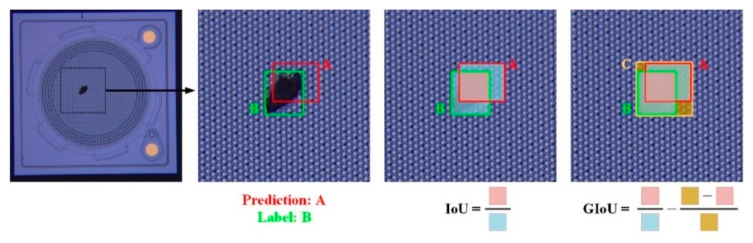
IoU and GIoU.

**Figure 6 sensors-22-09400-f006:**
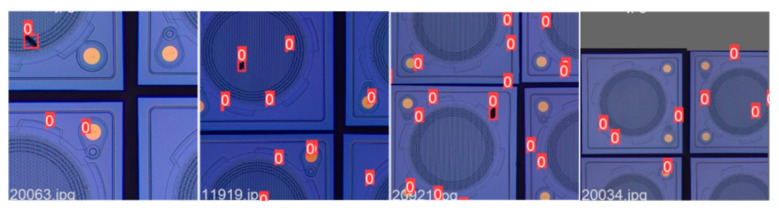
Mosaic data augmentation. (Red 0 represents single class label boxes.)

**Figure 7 sensors-22-09400-f007:**
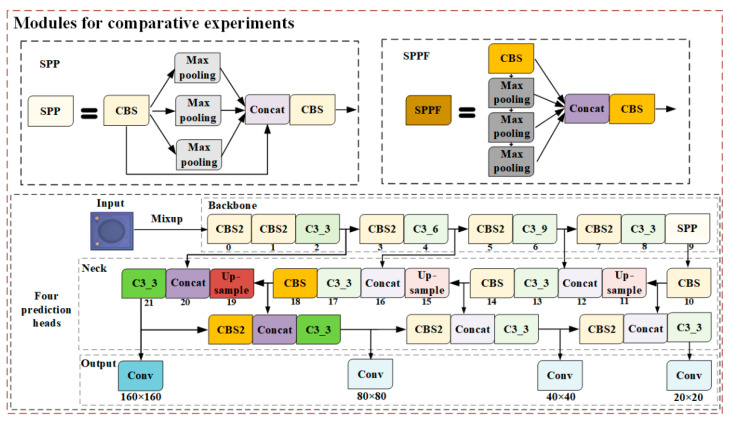
The modules for comparative experiments.

**Figure 8 sensors-22-09400-f008:**
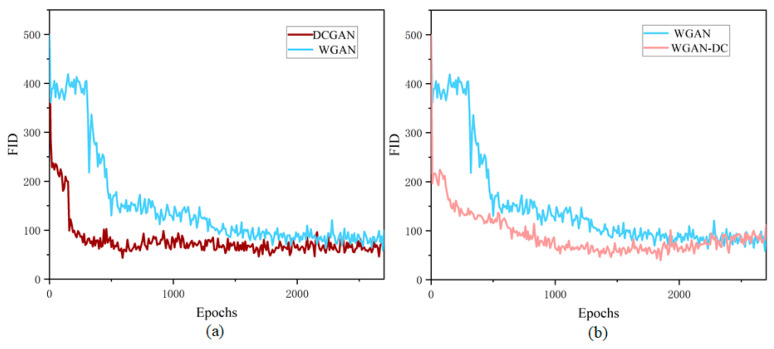
The comparison of FID curves: (**a**) DCGAN and WGAN; (**b**) WGAN and WGAN-DC.

**Figure 9 sensors-22-09400-f009:**
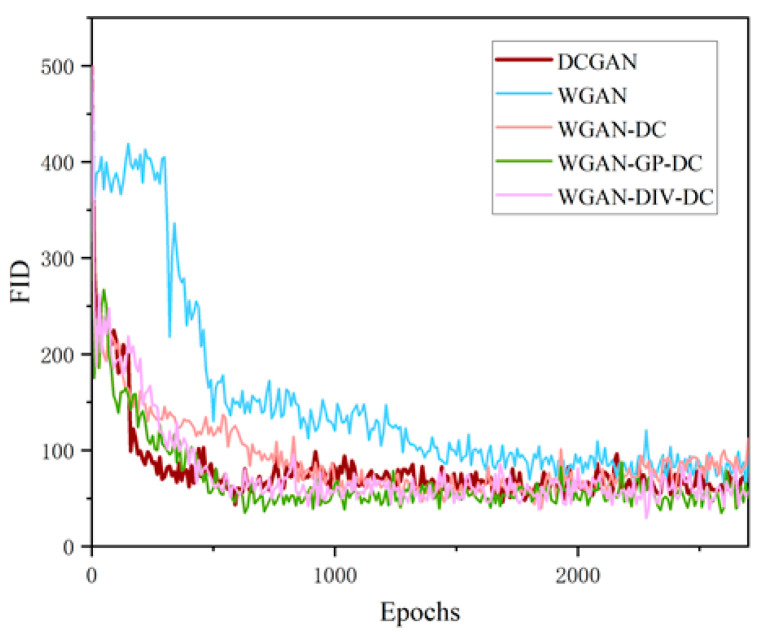
Comparison FID values of different GAN models.

**Figure 10 sensors-22-09400-f010:**
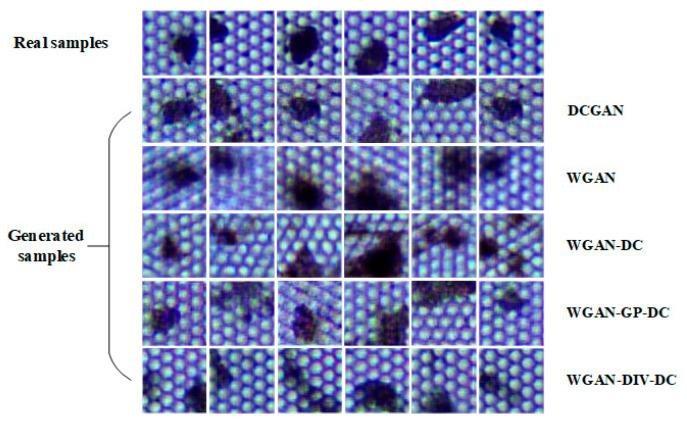
The real samples and the generated samples.

**Figure 11 sensors-22-09400-f011:**
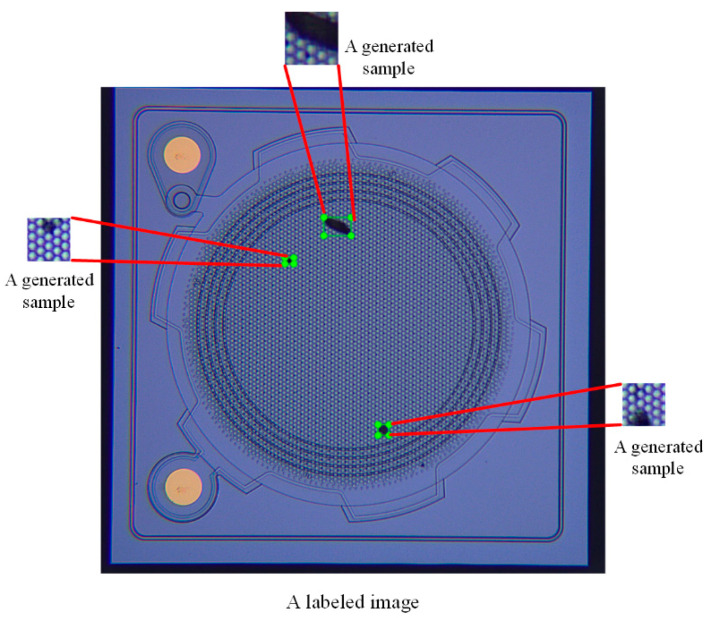
A synthetic image.

**Figure 12 sensors-22-09400-f012:**
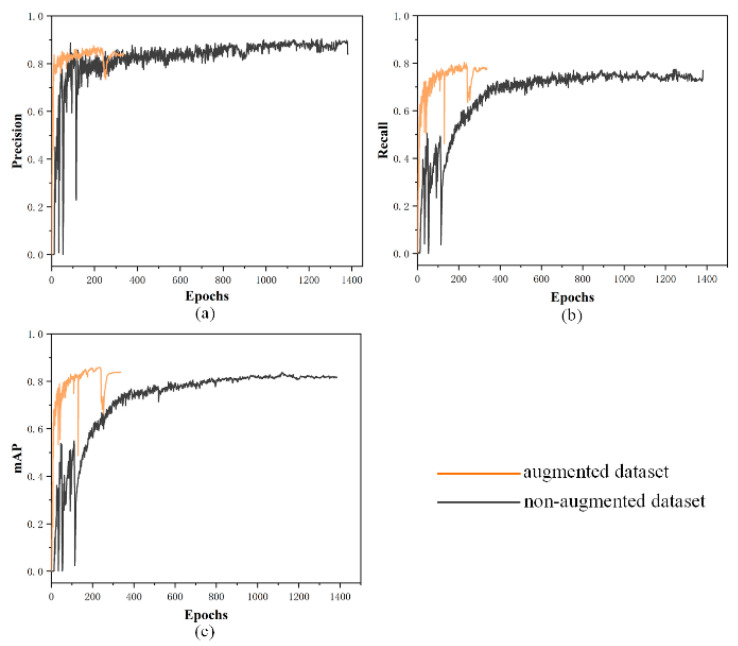
The variation curves of different metrics when the model was trained using an augmented dataset and a non-augmented dataset: (**a**) Precision curve of the models; (**b**) recall curve of the models; (**c**) *mAP* curve of the models.

**Figure 13 sensors-22-09400-f013:**
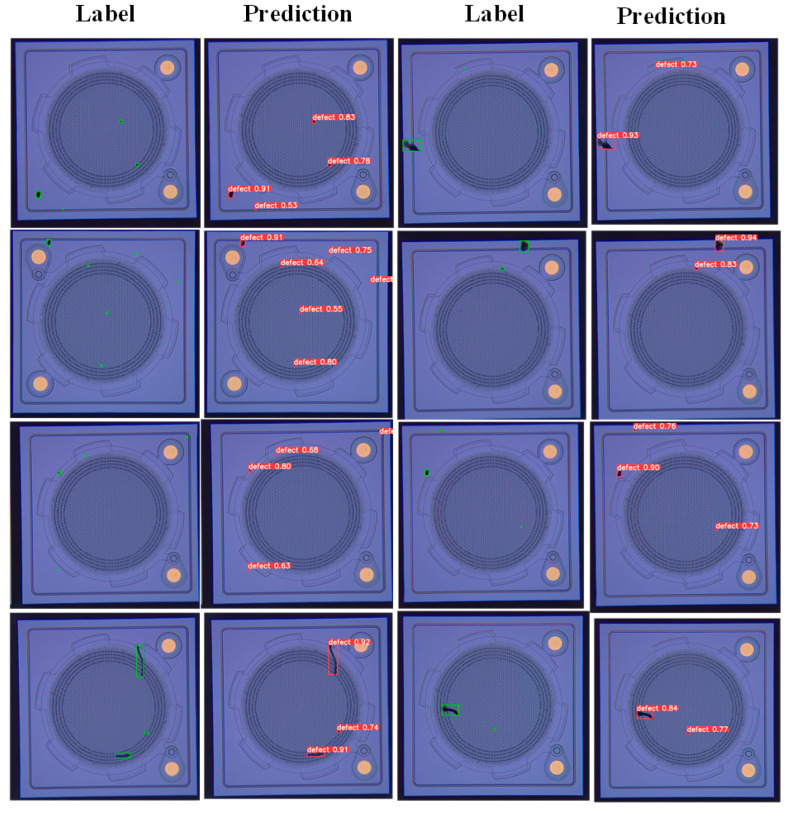
The detection results of the optimal model.

**Table 1 sensors-22-09400-t001:** FID score comparisons for different GAN models. The bold indicates the best model with the best performance.

Model	The Lowest FID
DCGAN	43.48
WGAN	54.47
WGAN-DC	39.55
WGAN-GP-DC	33.75
**WGAN-DIV-DC**	**29.44**

**Table 2 sensors-22-09400-t002:** The experimental datasets.

Data Augmentation	Train	Validation	Test	Total
Before	720	240	240	1200
After	4000	500	500	5000

**Table 3 sensors-22-09400-t003:** The experimental environment.

System	Ubuntu 18.04
CPU	Intel Core i7-9700f
GPU	2×NVDIA Geforce RTX 2080 Ti
Software	CUDA 10.1; Python 3.9; OpenCV 4.5
Framework	Pytorch 1.9

**Table 4 sensors-22-09400-t004:** The performance of the detection models.

	Baseline Model				The Optimal Model
Data augmentation with WGAN-DIV-DC		√	√	√	√
Mosaic			√	√	√
SPPF				√	
One more prediction head					√
Precision	0.858	0.871	0.873	0.884	0.865
Recall	0.752	0.78	0.817	0.807	0.847
*mAP*	0.833	0.867	0.889	0.888	0.901
F1 score	0.802	0.823	0.844	0.844	0.856

**Table 5 sensors-22-09400-t005:** Comparison between the SPP and SPPF.

Data augmentation	√	√
Mosaic	√	√
SPP	√	
SPPF		√
Total inference time	13.4 ms	13.3 ms
Detection speed	74.6 FPS	75.1 FPS

**Table 6 sensors-22-09400-t006:** Recent work comparisons.

Method	Ours	Improved YOLOv4 [29]	Faster R-CNN [14]
Object	MEMS surface defects	Concrete cracks	MEMS surface defects
*mAP*	0.901	0.925	
F1 score	0.856		0.81
Inference time/s	0.025	0.0279	18

## Data Availability

This study did not report any data.

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
