# Peer review of "Defect Detection of MEMS Based on Data Augmentation, WGAN-DIV-DC, and a YOLOv5 Model"

_sensors, 2022, doi:10.3390/s22239400_

Round 1

Reviewer 1 Report

The authors used an improved YOLOv5 to detect MEMS defects in real time. Added Mosaic and Four prediction heads to the YOLOv5 base model to improve feature extraction capability. To improve the detection accuracy of the YOLOv5 model, the number of error samples has been increased with Wasserstein deviation for producer adversary networks with deep convolutional structure (WGAN-DIV-DC). My reviews and suggestions about their publications are listed;

The article is generally well organized and prepared.

The "2. Datasets collection" section should be given as a subsection in one of the "3. Method" or "4. Experimental results and discussions" sections.

If possible, the related works section should be presented in the article.

Similar studies published in 2022 should be presented.

Although some evaluation criteria (Precision, Recall) are given in the article,  It should be well supported by Precision, Recall (sensitivity), Accuracy, Specificity, Prevalence, Kappa, and F1-score. These results need to be analyzed, tabulated, presented graphically, and interpreted.

Why did you use yolov5? YOLOv7, the newest YOLO algorithm surpasses all previous object detection models and YOLO versions in both speed and accuracy. It requires several times cheaper hardware than other neural networks and can be trained much faster on small datasets without any pre-trained weights. If possible, I suggest you try the YOLOv6 or YOLOv7 algorithms.

You should submit more experimental study results for your work. Sample applications and images should be presented on real images taken at different times. Experimental studies are insufficient. You should also provide comparisons with similar studies (In particular, studies in recent years).

Conclusion section should be extended. In other words, the author should discuss the future works related to proposed method and its drawbacks. Rewrite the conclusion with following comment: (a) Highlight your analysis and reflect only the important points for the whole paper. (b) Mention the implication in the last of this section. Please, carefully review the manuscript to resolve these issues. (c) This section should be supported with numerical values.

Author Response

      Thank you very much for helping us improve our manuscript in both scientific and linguistic aspects. We have uploaded the updated manuscript. Please see the attachment.

Reviewer 2 Report

Defect detection of MEMS based on data augmentation model WGAN-DIV-DC and YOLOv5  
The article in its current form lacks coherence in between discussions conducted in various sections. After a thorough study, the reviewer would like to highlight the followings:

Reviewer Comments:

1.      Add the following paper in the introduction section: “A Review Of Lattice Boltzmann Method Computational Domains For Micro- And Nanoregime Applications, Nanoscience and Technology: An International Journal, Vol. 11 (4), 343-373, 2020”

2.      Redraw Figure 2.

3.      Section 3.3 Title “Proposed improvements” can be changed.

4.      Section 4.1:  The specific parameters were set as follows: batch size to 64, training epochs to 2700, and optimized learning rate to 0.0002. Please clarify in the text.

5.      Redraw Figure 10.

6.      Conclusion part can be improved lot.

7.      Authors can provide Nomenclature.

 Revise and resubmit the paper. I recommend this paper to be published with the above changes indicated.    

Author Response

(The authors gave the same response as above.)

Round 2

Reviewer 1 Report

I have reviewed the revised manuscript title "Defect detection of MEMS based on data augmentation model WGAN-DIV-DC and YOLOv5". After revising my initial comments and comparing the changes, done by the authors, with them. I found that the authors addressed and answered most of the comments efficiently. Overall, the revised manuscrip is well organized and carefully prepared. The response letter was elegant and satisfactory. I thank the authors for their kind responses. The authors have sufficiently address my all comments. So, I think it is appropriate to accept the revised article. The authors have addressed all the concerns and responded to the review comments. The manuscript can be published in this journal.